# Where are you hiding the pangolins? screening tools to detect illicit contraband at international borders and their adaptability for illegal wildlife trafficking

**Georgia Kate Moloney** [1,2]*, **Anne-Lise Chaber** [1,2]

1 School of Animal and Veterinary Sciences, The University of Adelaide, Adelaide, SA, Australia, 2 Global One Health Alliance Pty Ltd, West Lakes Shore, SA, Australia

* georgia.moloney@adelaide.edu.au

**Data Availability Statement:** All relevant data are within the paper and its Supporting Information files.

## Abstract

The illegal movement of wildlife poses a public health, conservation and biosecurity threat, however there are currently minimal screening tools available at international ports of entry to intercept wildlife trafficking efforts. This review first aimed to explore the screening tools available or under development for the detection of concealed wildlife contraband at international ports, including postal services, airlines, road border crossings and maritime routes. Where evidence was deficient, publications detailing the use of methods to uncover other illicit substances, such as narcotics, weapons, human trafficking, explosives, radioactive materials, or special nuclear material, were compiled and assessed for their applicability to the detection of wildlife. The first search identified only four citations related to the detection of wildlife, however the secondary search revealed 145 publications, including 59 journal articles and 86 conference proceedings, describing screening tools for non-wildlife illicit contraband detection. The screening tools uncovered were analysed for potential fitness for purpose for wildlife contraband detection, to evaluate the feasibility of their implementation and their ease of use. The deficiencies evident in terms of resource availability and research efforts targeting wildlife trafficking highlights a potentially substantial national and international security threat which must be addressed.

## Introduction

Illegal wildlife trafficking is one of the greatest threats to international security and biosecurity, with estimates evaluating a trillion-dollar economic loss associated with the impact on biodiversity, ecosystem disruption and public health [1]. Wildlife is trafficked via various routes including postal services, airlines, road border crossings and maritime routes. Each year, over 159 billion packages, 22 million passenger flights and 800 million twenty-foot equivalent units (TEUs) of cargo containers are in transit worldwide, providing a multitude of opportunities for traffickers to smuggle wildlife products [2–4]. Many screening tools have been developed

**Funding:** This review was sponsored by World Wildlife Fund (WWF) France. Georgia Moloney received funding from the Australian Government Research Training Program Scholarship and the CMA CGM SA Supplementary Scholarship. The funders had no role in study design, data collection and analysis, decision to publish, or preparation of the manuscript.

**Competing interests:** The authors have declared that no competing interests exist.

for the detection of other illicit contraband transiting via these international routes, however there are significant shortcomings in the development or adaptation of tools for wildlife detection. Current limitations opposing adequate regulation include a lack of resources, funding and acknowledgement of the impact and consequences of the illegal wildlife trade.

The USAID Reducing Opportunities for Unlawful Transport of Endangered Species (ROUTES) Partnership was implemented between 2015 and 2021 to disrupt wildlife trafficking activities, with a particular focus on strengthening air transport routes. This initiative brought together transport and logistics companies, government agencies, development groups, law enforcement, conservation organizations, academia and donors, with a focus on improving data analytics, engaging corporate leadership, improving staff training to assist law enforcement, strengthening policies and protocols, and increasing collaboration between the transport sector and law enforcement [5]. Similarly, in 2022 the International Maritime Organization 'guidelines for the prevention and suppression of the smuggling of wildlife on ships engaged in international maritime traffic' were published to complement recommendations previously provided by relevant organisations to assist with reducing wildlife trafficking through the maritime sector [6]. These initiatives highlight the newfound acknowledgement of wildlife trafficking and the need for a multisectoral, collaborative approach. One of the deficiencies identified by these initiatives was the lack of screening tools available, but also a desire to improve these capabilities across sectors as the impacts of the illegal wildlife trade are recognised.

Wildlife trafficking is an all-encompassing term for the illegal movement of wild flora and fauna which can take many forms. Animals can be traded live or are poached for various parts including their bones, scales, meat, horns, skin, teeth, bile and blood. The raw materials themselves can be altered into powders, liquids, or other forms. These products are used for a wide range of applications including traditional medicines, clothing and accessories, while some also carry cultural or spiritual significance [7,8]. Plants and their derivatives including timber, seeds, orchids, oils, succulents and other products are also traded, wherein certain species are highly valued for their cultural, medicinal, or ornamental properties [9]. Due to this endless catalogue of products, there is a need for different screening tools to uncover these commodities based on their inherent properties.

With enhanced accessibility to the sale of wildlife products and an increasingly interconnected society, additional safeguards must be implemented. Without the installation of efficient screening tools at international ports, wildlife traffickers will continue to evade authorities with potentially devastating consequences. Therefore, we aimed to investigate the scientific literature for tools and technologies which are currently available for the detection of illegally trafficked wildlife contraband at international ports, including postal services, airports, vehicle border checkpoints and shipping ports. Where no publications are available, literature describing tools either developed or investigated for the detection of other illicit contraband will be screened and the potential for adaptation to the detection of wildlife will be assessed. We aim to provide a practical guide for law enforcement agencies (LEA) to strengthen their knowledge and capabilities and to raise awareness for the significance of the illegal wildlife trade.

## Methods

### Data sources and search strategy

The scoping review was conducted in accordance with the Joanna Briggs Institute methodology for scoping reviews [10] and Preferred Reporting Items for Systematic Reviews and Meta-Analyses (PRISMA) Extension for Scoping Reviews Checklist [11] to ensure a thorough and

well-informed analysis of the available literature. A preliminary search of review databases was conducted and the authors identified no current or underway scoping or systematic reviews summarising wildlife detection tools. The following questions were used to guide this review: (i) What tools (physical screening tools only) are currently being implemented to detect illegal wildlife products at airports, vehicle border checkpoints, shipping ports and postal facilities and how effective are these methods? (ii) What additional tools exist to enable the detection of other illicit contraband at ports of entry? (iii) How can these techniques be applied to wildlife contraband detection? The search strategy aimed to uncover primary scientific reports, reviews and conference proceedings published between 1990 and 2023. The initial literature search was conducted between 28 February and 8 March 2022 across four databases including PubMed, CAB Abstracts, Web of Science and Google Scholar. An additional literature search was conducted in IEEE Xplore and Scopus on 4 April 2023. The protocol adapted for this review to screen articles consisted of three levels: (1) screening based on title and abstract, (2) screening of the full article, and (3) data extraction from selected articles.

## Current tools for wildlife trafficking detection

Published literature was first assessed for evidence of detection tools specifically being investigated or implemented to combat wildlife trafficking. Combinations of key terms relevant to the review aims were trialled, consisting of 'illegal wildlife trade' synonyms paired with detection methods and/or known modes of transport. The full search strategy is outlined in the Supporting Information (S1 Table). Only articles referring directly to a method of detection and containing a combination of select search terms in the title, abstract or key words were considered for analysis. Meanwhile, articles wherein the full text was inaccessible, or which referred to the identification of a sample which had already been seized (as opposed to the initial detection of the product) were excluded. For databases which generated thousands of results, the *a priori* decision was made to only screen the first 200 articles. While this approach was elected to expedite the initial screening process, we acknowledge that it may have potential consequences for our empirical results, but believe this to be balanced by our subsequent rigorous screening and selection processes. Only four articles detailing wildlife detection methods were uncovered [12–15]. The reference lists of these papers were screened for additional resources but generated no suitable articles. Whilst limited papers were discovered through the search strategy employed, the authors are aware of other technologies being trialled for wildlife detection, such as environmental DNA (eDNA) [16–18], however they are either not currently used or evaluated in the context of wildlife contraband screening.

## Current tools for other illicit contraband detection

Due to the substantial lack of literature pertaining to the detection of illegal wildlife products, published literature detailing screening tools and technologies for other illicit contraband were explored. A preliminary search of the internet and grey literature was conducted for key terms to guide the database search term generation. The database search centred around major international crimes including the transport of illegal narcotics, weapons and human trafficking, with the addition of detection methods and transport routes. The full search strategy is provided in the Supporting Information (S2 Table). Citations were imported into EndNote 20 (v20.2.1) [19] and duplicates were removed. Titles and abstracts were screened by the reviewers for assessment against the review inclusion criteria. Articles were included if they detailed a detection method used to uncover illicit contraband such as narcotics, weapons, human trafficking, explosives, radioactive materials or special nuclear material. Potentially relevant sources and their citation details were retrieved in full for further analysis. Sources were

excluded at this stage if the full text was inaccessible, or if they described the identification of a type of contraband (i.e. after it has been detected) as opposed to screening for the unknown presence of the product. Reasons for exclusion of sources which did not meet the inclusion criteria have been reported. For example, cases uncovering 'body packers' (i.e. persons who have swallowed containers of narcotics or other illicit substances to conceal them) or 'body pushers' (i.e. persons who conceal containers of illicit substances in their vagina or rectum) were excluded as these techniques were not considered relevant to wildlife trafficking [20]. The reference list of all included articles were also scrutinised for additional references detailing novel screening methods or applications. Metadata was extracted and collated from selected articles (Supplementary Information). The overall search and inclusion process is presented in Fig 1.

## Results

A total of 145 articles were included in the study consisting of 59 peer-reviewed scientific articles and 86 conference proceedings. Of the published articles, 39% were conducted in or sponsored by the United States of America (23/59), followed by 19% in the Netherlands (11/59) and 15% in the United Kingdom (9/59). Meanwhile, 55% of the conferences occurred in the United States of America (47/86), followed by 26% in Europe (22/86) and 17% in Asia (15/86).

We identified 59 different inspection systems or system variations (including artificial intelligence (AI)) and categorised them for ease of reader comprehension across six categories based on their mode of action or required sample type: ionising radiation, photon/neutron interrogation, nuclear, non-ionising radiation, trace detection and acoustic (S3–S8 Tables). Detection methods were applicable for a wide variety of contraband including narcotics and other drugs (such as cigarettes), weapons, human trafficking, nuclear waste, special nuclear material, explosives, chemical warfare agents, organic and inorganic materials, radiological material and other hazardous substances. The relative proportion of the literature associated with each category is represented in Fig 2.

### 1. Ionising radiation

Techniques utilising ionising radiation are the most widely used for contraband detection, with 31 papers describing 11 different ionising radiation tools (including AI enhancements) currently in use (S3 Table). The techniques described enable material discrimination and determination of the internal properties and contents of objects, including shape, density, location, relative atomic number and effective nuclear charge ($Z_{eff}$). Ionising radiation scanning systems are non-invasive, versatile and have been deployed in various environments. One of the main disadvantages with these systems however is the health and safety concerns for personnel, as ionising radiation leads to the formation of oxygen-derived free radicals which damage cellular DNA [21]. Radiation sources are controlled worldwide by International Atomic Energy Agency safety standards, under which all radiation sources must be registered and companies and users are required to hold a license [22]. The two main ionising radiation systems described were x-ray radiography (including single view, dual angle and backscatter variations) and computed tomography (CT), meanwhile gamma radiation or passive radiation detection may be employed in specific circumstances to identify the elemental composition of materials. In addition, there were five different AI programs described to enhance digital imaging processing techniques, including automatic image recognition to assist in the identification of weapons, explosives and other dangerous goods for x-ray images of luggage or cargo [23–33]. AI has been described as a powerful addition to these systems for enhanced efficiency and adaptability to other types of contraband.

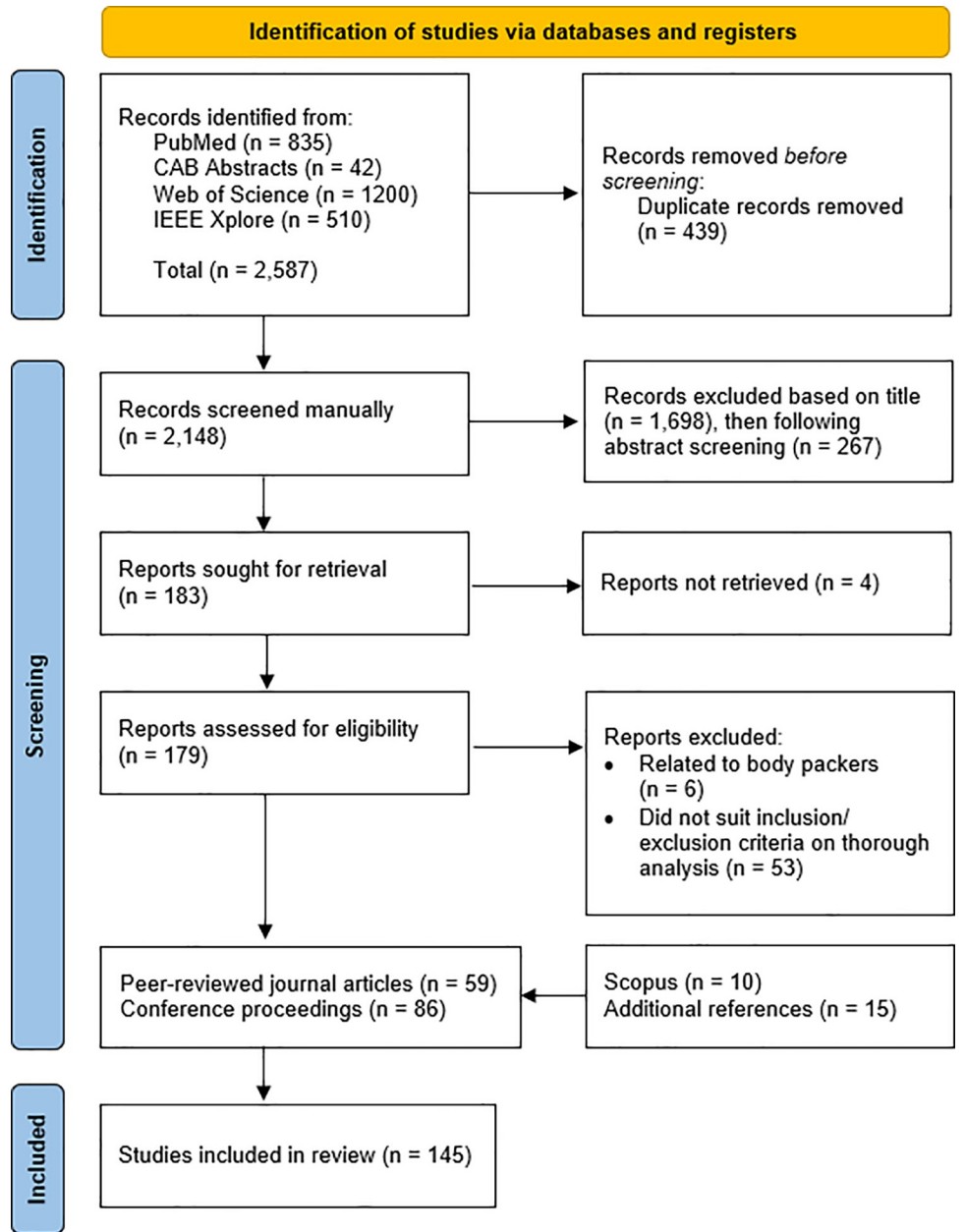

**Fig 1. Review methodology flowchart.** Preferred Reporting Items for Systematic Reviews and Meta-analyses extension for scoping review (PRISMA-ScR) flowchart illustrating the search strategy and screening process for articles published between 1990 and 2023. The search criteria identified 2,587 articles, which were then refined as per the process described in the flowchart to produce 145 English language full-text articles for analysis. A further fifteen references describing novel techniques or applications derived from reference lists (snowballing) were also included.

## 2. Photon and neutron interrogation techniques

Photon and neutron interrogation techniques exploited in 11 tools were described in 25 papers and rely on the determination of the elemental composition or ratio of interrogated materials to support contraband identification (S4 Table). Ionising radiation is used in these methods, where most employ pulsed or fast, thermal or non-thermal, neutrons or photons [34–41]. Fast neutron and high-energy gamma-ray scanners have been explored in combination with x-ray

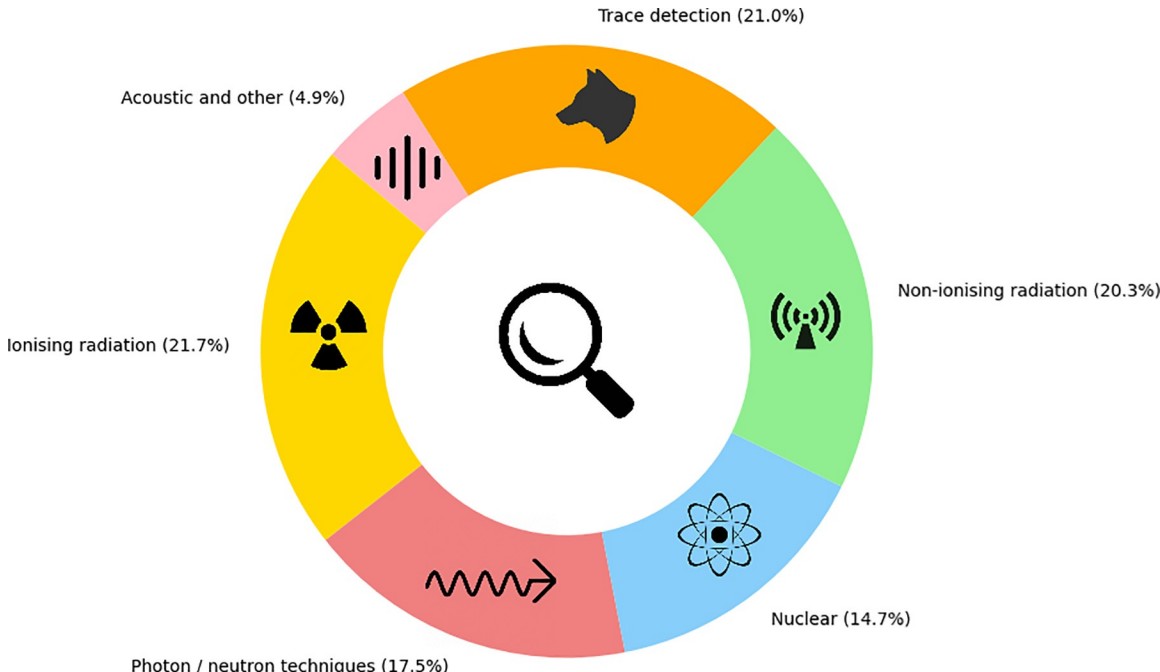

**Fig 2. Distribution of literature analysed.** The proportion of literature cited describing detection tools within each of the six categories based on their mode of action or required sample type.

radiography to enable the determination of compositional information and discrimination between a wide range of organic materials. The combination of these techniques enables the detection of narcotics, chemical, biological, radiological, nuclear and explosive (CBRN&E) materials [42]. While organic materials may be detectable through these methods, their properties are largely not conducive to detecting wildlife products and thus they are unlikely to be useful for this application, hence they will not be discussed in further detail.

### 3. Nuclear material detection systems

A total of 21 papers describing eight different nuclear-based detection systems were included in this review (S5 Table). Nuclear-based methods compare elemental information for material detection, such as nitrogen content for explosives [43]. Some techniques derive both images and elemental information based on the irradiation of an object with thermalised neutrons and sensitive detection of the associated/scattered gamma-rays produced for the detection of special nuclear material, narcotics and explosives concealed in cargo or luggage [44,45]. However, as there are no known examples of these technologies being used to detect wildlife or similar contraband, they are unlikely to be useful for this application, hence will not be discussed in further detail in this report.

### 4. Non-ionising radiation

Non-ionising radiation techniques, such as millimetre wave and terahertz scanners, are also popular and avoid the safety concerns associated with ionising radiation. In this review, seven different tools were described across 29 papers (S6 Table). These systems are considered safe, non-intrusive and offer full-body screening while preserving the privacy of individuals by generating images that reveal concealed items but not anatomical details. Harmer et al. even described the application of millimetre wavelengths to image footwear without necessitating

prior removal, which would greatly accelerate passenger screening processes [46]. Their ability to deliver rapid, real-time results contributes to efficient screening processes in high-traffic areas. Furthermore, their versatility and ability to penetrate clothing and non-metallic materials allows for the detection of various contraband types, including weapons, explosives and narcotics [47,48]. These systems can be integrated into existing security infrastructure, offering adaptability and complementing other screening technologies. However, non-ionizing radiation systems may struggle to distinguish between specific materials or chemical compositions, potentially affecting their effectiveness in identifying certain contraband types. The interpretation of images can be complex, necessitating well-trained operators and, while designed to respect privacy, concerns about data collection and usage may persist. These technologies have been widely used in airports and postal facilities worldwide and complement current security procedures.

## 5. Trace detection techniques

Trace detection refers to the detection of minute substances which cannot be seen with the naked eye that may indicate the presence of illegal or harmful materials, such as particles on surfaces, odours or volatile organic compounds (VOCs). Across the 30 papers focusing on trace detection techniques included in this review, 19 different tools were described (S7 Table). These systems enable fast, non-invasive and sensitive screening, but typically work best in a controlled environment with minimal airflow [49–51]. Most methods are highly sensitive, minimally invasive, rapid, safe, maintain sample integrity and require minimal or no sample preparation [52,53]. Most require the detection of a specific biomarker, such as VOCs, and can identify a range of contraband including CBRN&E materials and even animals. However, while trace detection systems are sensitive, their ability to identify specific substances or materials is limited, often requiring additional analysis for confirmation. False positives can also occur due to the presence of common substances such as medications. The maintenance and calibration of these systems is critical for optimal performance and their effectiveness is contingent on operator training and vigilance.

## 6. Acoustic and other

The remaining seven papers described three acoustic and related technologies (S8 Table). Acoustic or ultrasonic pulses can be used to determine the physical properties of a material and feature advantages over x-ray inspection systems as they are able to penetrate dense materials and liquids. Pulses are launched into a material or container, then the returning echoes are analysed based on frequency and time-of-flight to determine the physical properties of the object, namely acoustic velocity and attenuation coefficient. These methods have been used to uncover concealed explosives and other contraband in containers [54–57]. Respiration of concealed persons in cargo containers can also be determined using this method through measurement of time-varying phase shift of the returning acoustic wave using high-power ultrasonic transducers [57]. Smart containers are another innovation currently being commercialised to enable real-time monitoring of container status and cargo contents. These containers are fitted with specialised sensors to collect and transmit information directly to operators to enable constant monitoring of cargo and provide non-specific indicators of suspicious activity [58,59]. This technology is listed in this category for the purposes of this report as acoustic sensors have been trialled in smart containers to identify sounds which may be exclusively associated with an attack (i.e. a grinder) to reduce false alarm rates [59].

## 7. Applicability to wildlife trafficking

Based on the properties of the tools explored through this review, those most likely to be adapted for wildlife trafficking detection have been identified in Table 1, listed with their main advantages as well as limitations and shortcomings. We have broadly grouped these into categories (for example, while there were seven different x-ray variations, they have been generally classed as 'x-ray') to improve the clarity of this manuscript. While these tools may be appropriate for wildlife detection based on their mode of operation, it is also important to consider their inherent properties and whether they are ideally suited to law enforcement operations [60]. Many of the properties considered important when selecting a detection tool have been evaluated with the tools and should also be taken into consideration (Table 2).

# Discussion

## Summary of the literature

The findings of this review highlight tools and techniques currently in use or under investigation to facilitate contraband detection at international borders and ports. A total of 59 screening tools for the detection of illicit contraband at airports, vehicle border crossings, postal services and maritime ports were described in the literature assessed. However, interestingly there was a difference between the technologies available across these different sectors, not necessarily proportional to the volume of goods handled or known contraband smuggling routes. More than half of the technologies described were available for airports, including air freight, luggage and personnel screening, particularly targeted towards uncovering illicit weapons and drugs. The expansion of these technologies demonstrates a reactionary approach to risk mitigation, with a significant increase in research and security efforts particularly evident after the infamous September 11 terrorist attacks in the United States of America. The distribution of research also closely reflects government priorities, with the United States driving airport security research and implementation, reflected by 39% and 55% of publications and conference proceedings respectively. However, these events should highlight the importance of proactive measures to prevent the devastating consequences associated with illicit activities. The international illegal wildlife trade poses a major threat to biodiversity and public health [107] and has even been linked to recent global pandemic events [108]. However, screening tools used for wildlife trafficking detection were severely lacking in the scientific literature. Currently, wildlife seizures predominately rely on prior intelligence as opposed to active surveillance methods, thus seizures reported likely represent a very small percentage of all smuggling attempts. The aim of implementing screening tools is to increase the probability of detection of wildlife trafficking, even in the absence of intelligence.

## Challenges for screening tool implementation

All stakeholders and international ports share similar requirements for screening tools, including compatibility and adaptability, versatility, non-invasiveness, automation ability, efficiency, sensitivity, specificity, portability and ease of operation and sample collection [60]. However, screening requirements can vary greatly between countries and ports due to local regulations, political will, infrastructure, resource availability and threat perception, which can be influenced by cultural and political differences and prior security threats encountered [8]. Ideally, standardised screening protocols should be implemented to ensure consistency and predictability. Technologies available in some regions are restricted due to resource limitations and each country is at a different stage in fighting wildlife crime [8]. Unfortunately, traffickers will leverage any known weaknesses in the supply chain. For example, if traffickers were seeking to

**Table 1. Summary of the detection tools deemed applicable for wildlife trafficking detection, with their main benefits and limitations as described in the literature.**

| Inspection system | Description | Benefits | Limitations | References |
|---|---|---|---|---|
| *Ionising radiation* | | | | |
| X-ray | High-energy x-rays provide a powerful imaging tool used in contraband detection. These x-rays can penetrate dense materials and provide detailed scans as they pass through an object to produce a high-resolution 2D image, making them essential for uncovering hidden contraband, weapons, or illicit items in cargo, luggage, or vehicles. Imaging processing methods are employed for enhanced material discrimination. | ✓ High resolution images<br>✓ Improved x-ray penetration<br>✓ Good discrimination between metals and organic materials<br>✓ Reduced false alarm rates and enhanced detection sensitivity<br>✓ Beam collimator reduces unwanted scattered radiation<br>✓ Portable systems available<br>✓ Versatile and adaptable for various applications<br>✓ Non-invasive<br>✓ Non-destructive | ✗ AI required to improve material discrimination capability<br>✗ 2D imaging<br>✗ Fixed/large x-ray systems for container scanning<br>✗ Radiation safety concerns<br>✗ Specialised training required for staff to interpret images<br>✗ Ongoing compliance requirements for operation | [61–74] |
| Computed Tomography (CT) | CT is used in contraband screening to create detailed 3D cross-sectional images of cargo, luggage or vehicles using high-energy x-rays, allowing inspectors to identify various types of concealed contraband. | ✓ 3D imaging<br>✓ Provides quantitative data<br>✓ High resolution images<br>✓ Material discrimination<br>✓ Detailed imaging<br>✓ Detection of contraband is independent of its position and orientation<br>✓ Non-invasive<br>✓ Non-destructive<br>✓ No unpacking required<br>✓ Reduced false alarm rates<br>✓ Can be automated | .✗ Radiation safety concerns<br>✗ Large fixed system<br>✗ More expensive than x-ray systems<br>✗ Requires trained operators | [62,63,75–77] |
| Artificial intelligence (AI) | AI is revolutionizing contraband detection by enhancing the speed, accuracy, and consistency of screening processes. AI systems analyze data from various sources, including x-ray scanners, millimeter-wave scanners and thermal imaging, to identify hidden contraband items using pattern recognition to detect anomalies that may elude human inspectors. AI's ability to rapidly process and interpret vast amounts of data makes it a valuable tool in improving security and efficiency at customs checkpoints, border crossings and cargo screening facilities. | ✓ Non-invasive<br>✓ Enables automation of scanners<br>✓ Uses existing infrastructure<br>✓ Can process large volumes of data or images efficiently<br>✓ Increases workload capabilities<br>✓ Improves detection rates<br>✓ Maintains consistency<br>✓ Updated as new research is undertaken<br>✓ Does not interfere with safety screening<br>✓ Minimal training required for use<br>✓ Complementary to human operators | .✗ Requires extensive training data<br>✗ Potential privacy concerns<br>✗ Regulations and legal frameworks governing the use of AI in security and customs screening may vary by jurisdiction<br>✗ Requires human oversight for decision-making<br>✗ Possibility for false positives<br>✗ Currently not available for shipping container screening<br>✗ Not all ports have existing infrastructure compatible with AI programs | [23–33] |
| *Non-ionising radiation* | | | | |
| Millimeter wave imaging | Millimeter-wave imaging is a whole-body imaging technology. Active methods emit low-power millimeter waves and measure their reflections to create high-resolution 3D images. This technique can reveal concealed items, such as weapons or drugs, on a person's body, shoes, or within objects. Passive millimeter-wave imaging captures naturally emitted millimeter-wave radiation from objects and individuals. Image fusion technology (combining a visual image with a millimeter-wave image) can improve detection efficiency. | ✓ Non-ionising radiation<br>✓ Whole-body scanning<br>✓ Detection of concealed objects on persons<br>✓ 3D imaging<br>✓ Non-invasive<br>✓ Non-destructive<br>✓ Rapid screening<br>✓ Can detect contraband in shoes, even while being worn<br>✓ Active or passive variations | .✗ Minimally detailed images<br>✗ Limited spatial resolution<br>✗ Less effective at detecting non-metallic items (i.e. certain explosives)<br>✗ Requires regular maintenance<br>✗ Limited depth of detection (primarily surface scanners)<br>✗ Short range<br>✗ Potential privacy concerns | [46–48,54,78–82] |

*(Continued)*

**Table 1.** (Continued)

| Inspection system | Description | Benefits | Limitations | References |
|---|---|---|---|---|
| **Terahertz spectrometry** | Terahertz spectrometry operates within the terahertz frequency range to analyze the unique spectral signatures of objects or materials. Passive terahertz systems utilise naturally emitted radiation by the human body to detect hidden objects (i.e. weapons hidden under clothing). | ✓ Non-invasive<br>✓ Non-ionising radiation<br>✓ High resolution images<br>✓ Whole-body scanning<br>✓ Detection of concealed objects on persons<br>✓ Identification of liquids and powders<br>✓ Non-destructive<br>✓ Active or passive variations<br>✓ Material discrimination<br>✓ Remote sensing applications | ✗ Limited penetration, particularly in dense or opaque materials<br>✗ Possible false alarms when scanning objects with similar THz spectral characteristics<br>✗ May be affected by environmental factors, such as temperature and humidity<br>✗ Potential privacy concerns | [83–92] |
| **Infrared imaging (IR)** | Infrared imaging captures thermal radiation emitted by the body or objects to form an image. Through identifying variations in temperature, it can reveal hidden items within objects or carried by individuals. The relative temperature of concealed objects and the human body are measured. Short-wave infrared variations use electromagnetic radiation, where photons are reflected or absorbed by the object and interpreted to produce an image. | ✓ Non-invasive<br>✓ Non-ionising radiation<br>✓ Minimal false detections<br>✓ Hand-held<br>✓ Real-time results<br>✓ Detection of concealed objects on persons<br>✓ Low-light operation<br>✓ Contactless, distance screening | ✗ Relies on temperature differences<br>✗ Visibility of concealed objects decreases as the temperature difference between the object and the human body decreases<br>✗ Limited material discrimination<br>✗ Affected by environmental conditions<br>✗ Potential privacy concerns | [85,93–95] |
| *Trace detection* | | | | |
| **Detector dogs** | Detection dogs are highly trained canines used to identify a wide range of contraband. Their exceptional sense of smell enables them to detect hidden items with remarkable accuracy. | ✓ Non-invasive<br>✓ Portable and mobile<br>✓ High sensitivity<br>✓ Can screen large quantities/areas quickly<br>✓ Can be trained to detect multiple different scents, including wildlife-specific scents<br>✓ Act as a force-multiplier<br>✓ Provide a physical presence (deterrent)<br>✓ Can be trained to accurately detect a range of contraband types | ✗ Ongoing training, veterinary and husbandry requirements<br>✗ Affected by environmental conditions<br>✗ Requires a supply of suitable dogs<br>✗ Operational time relies on dogs' motivation and ability to continue working<br>✗ Relies on the detection of odours, which may not always be present or distinguishable<br>✗ Sensitivity obscured by concealment techniques or background odours<br>✗ Limited standoff distance (need proximity) | [14,96] |
| **Electronic nose (eNose)** | The eNose is a contraband detection device that mimics olfaction. It uses sensors to detect and analyze chemical odours emitted by substances such as explosives or drugs. The detector is composed of a sampler and an analyser, to first extract vapours then analyse them. | ✓ Non-invasive<br>✓ Non-destructive<br>✓ Portable systems available<br>✓ Real-time data<br>✓ Minimal staff training required<br>✓ Can sample from a distance<br>✓ Automation<br>✓ High sensitivity<br>✓ Multiple odour recognition | ✗ Affected by environmental conditions<br>✗ Interference from background scents or contraband concealment methods<br>✗ Limited specificity between similar odours<br>✗ Requires regular maintenance and calibration<br>✗ Lack of standardised procedures for implementation | [50,51,97–102] |
| **Mass spectrometry (MS)** | MS is a powerful tool in contraband screening operations. It enables the detection and identification of a wide range of illicit substances, including drugs, explosives and chemicals, with high sensitivity and accuracy. MS provides rapid, quantitative analysis and detailed information about the composition and quantity of contraband materials, thus aiding security personnel in threat assessment and decision-making. Various tools using MS variations were described in the literature. | ✓ Non-invasive<br>✓ High sensitivity<br>✓ Accurate identification of substances<br>✓ Can analyse a wide range of analytes and sample types/mediums<br>✓ Quantitative analysis<br>✓ Enables the development of a library of analytes which can be used for identification by eNose | ✗ Requires highly trained operators<br>✗ Large systems are not portable<br>✗ Expensive and highly specialised facilities required<br>✗ Long data acquisition times possible<br>✗ High-quality mass spectrometers can be expensive to operate and maintain<br>✗ Sample preparation required<br>✗ Background environmental interference possible | [49,103–106] |
| *Acoustic and other* | | | | |

(*Continued*)

**Table 1.** (Continued)

| Inspection system | Description | Benefits | Limitations | References |
|---|---|---|---|---|
| **Smart containers** | Smart containers are equipped with sensors and communication technologies to continuously monitor cargo conditions in real-time. They detect GPS positioning, acceleration, door-opening or tampering, and irregularities such as unauthorized access or changes in humidity and temperature. Additional sensors may also be equipped in the container. | ✔ Non-invasive<br>✔ Real-time monitoring<br>✔ Enhanced control over shipments<br>✔ Cargo condition monitoring<br>✔ Security alerts<br>✔ Inventory management<br>✔ Battery life of 5–8 years<br>✔ Automated<br>✔ Improves safety and customer confidence | ✗ Non-specific<br>✗ Not all containers are equipped with this technology<br>✗ Additional cost to shipping companies<br>✗ Data privacy concerns<br>✗ Security vulnerability if hacked | [58,59] |

smuggle products from Africa to Europe via airports, they may take advantage of an initial connection to another African country with weaker screening technology, therefore avoiding the more advanced technology at the European destination as baggage is not re-screened on arrival [109]. Where possible, research efforts should be focused towards underdeveloped regions with limited resource capacity, especially as these regions are more frequently associated with wildlife trade activities. Different transport modalities also handle goods differently and in diverse quantities. For example, postal facilities handle thousands of letters and packages daily, but each item contains a relatively small quantity, in comparison to shipping ports which handle thousands of containers and much larger quantities of goods. The time taken for transport chains to deliver goods also varies, where for example in the context of the wildlife trade traffickers are more likely to smuggle live animals either in the mail or via airlines as transit times are much shorter. Comparatively, maritime routes or trucks are more likely to be used where large quantities of wildlife products (from deceased animals or plant derivatives) are transported between international or land-locked countries, respectively. While passenger luggage is largely screened at the point of loading, most luggage, parcels and containers are only re-screened upon arrival into a country for biosecurity concerns, with varying entry requirements and procedures implemented between countries. In the context of the wildlife trade, it is important to consider the differences in routes and methods used to tailor screening efforts. A combined bulk detection method (used to determine the size and shape of suspicious objects) and trace detection method (detects trace contaminants and identifies chemicals, with higher selectivity than bulk detection) may be ideal to improve confidence. Screening tools demonstrating promise for adaption for wildlife detection are described using case-based scenarios, but these technologies are not exclusive to one application or port and may be used in various scenarios across these sectors.

## Screening of passenger luggage through airports

Techniques utilising ionising radiation are commonly employed in ports of entry, thus the infrastructure is widely accessible and suitable for border control operations. They are non-invasive and easy to use, offering either single or dual views in pseudo-colours which facilitate material differentiation [110,111]. X-ray systems including single view, dual view and back-scatter scanner variations are appropriate for investigating animal anatomy, particularly bones, horns, ivory and teeth, but also to enable the detection of other products through the distinction of organic materials. Media reports have exposed the use of x-ray scanners to uncover wildlife trafficking attempts in passenger luggage, such as in 2020 where two women were caught smuggling 109 live animals from Thailand to India [112]. CT scanners are an

**Table 2. Summary of the ideal properties of the detection tools deemed applicable for wildlife trafficking detection.** Note that this summary is intended to be used as a visual aid and guide only. An increase in the number of symbols associated with each category indicates how strongly it applies to the detection tool, unless otherwise indicated.

| | Transport sector[a] | Start-up costs[b] | Operational costs[c] | Time to implement | Adaptability | Versatility | Portability | Efficiency | Staff requirements |
|---|---|---|---|---|---|---|---|---|---|
| X-ray 🐾 | 👤🧳📦🚜📦 | €€ | €€ | 🕐 | ⚙️⚙️ | ❄️❄️❄️ | 📟📟📟 | ⚙️⚙️⚙️ | 👥👥 |
| Computed tomography 🐾 | 👤🧳📦🚜📦 | €€€ | €€ | 🕐 | ⚙️⚙️⚙️ | ❄️❄️❄️ | 📟 | ⚙️⚙️⚙️ | 👥👥 |
| Artificial intelligence* 🐾 | 👤🧳📦🚜📦 | € | € | 🕐 | ⚙️⚙️⚙️ | ❄️❄️❄️ | 📟📟📟 | ⚙️⚙️⚙️ | 👤 |
| Millimeter wave imaging | 👤🧳📦🚜📦 | €€ | €€ | 🕐 | ⚙️⚙️ | ❄️❄️ | 📟 | ⚙️⚙️⚙️ | 👤 |
| Terahertz spectrometry | 👤🧳📦🚜📦 | €€ | €€ | 🕐 | ⚙️⚙️ | ❄️❄️ | 📟📟 | ⚙️⚙️⚙️ | 👤 |
| Infrared imaging | 👤🧳📦🚜📦 | € | € | 🕐 | ⚙️⚙️⚙️ | ❄️ | 📟📟📟 | ⚙️⚙️⚙️ | 👤 |
| Detector dogs 🐾 | 👤🧳📦🚜📦 | €€€ | € | 🕐🕐🕐 | ⚙️⚙️⚙️ | ❄️❄️❄️ | 📟📟📟 | ⚙️⚙️⚙️ | 👥👥👥 |
| Electronic nose 🐾 | 👤🧳📦🚜📦 | € | € | 🕐 | ⚙️⚙️ | ❄️❄️❄️ | 📟📟📟 | ⚙️⚙️ | 👤 |
| Mass spectrometry | 👤🧳📦🚜📦 | €€ | €€ | 🕐 | ⚙️⚙️ | ❄️❄️❄️ | 📟📟 | ⚙️⚙️ | 👥👥 |
| Smart containers | 👤🧳📦🚜📦 | € | € | 🕐🕐 | ⚙️⚙️⚙️ | ❄️❄️❄️ | 📟📟📟 | ⚙️⚙️⚙️ | 👤 |

[a] Transport sector or goods type the technology could be applicable to, represented by the human body, luggage, postal/courier parcel, road or air cargo shipment and containerised cargo (in order of symbols). Green indicates the applicable sectors, whereas grey represents where these transport modalities do not apply.

[b] Estimated costs as of June 2023 in euros associated with the initial purchase and set up of the equipment, based on grey literature and personal contact with airport, port and law enforcement authorities. Prices are always subject to variation; this table is intended to provide an indication of costs in the context of the review only. € up to 50k, €€ up to 500k, €€€ up to 1 million euros.

[c] Estimated costs as of June 2023 in Euros for one machine, piece of equipment or one trained detector dog.

*Can vary considerably depending on the source. Many are free, low cost or are already programmed into the machines. Alternatively, some companies may be working on their own AI models.

🐾 Have already been used or are currently employed to detect wildlife.

advanced screening technology favoured over conventional x-ray systems to produce high resolution 3D images of cargo or luggage with the advantages of automation, ability to characterise more complex objects and providing positioning information. However, these systems are not portable or hand-held, are typically more expensive than 2D x-ray scanners and some units may require a longer scanning time, which may present resource and capacity limitations in at-risk countries. CT scanners are already required by law to screen hold baggage in the United States and Europe and are even being used in some Asian and African regions. While CT systems are adaptable, they are non-specific and require trained operators to classify objects. In areas with limited access to CT scanners, conventional x-ray scanners can be used,

but may provide comparative weaknesses in the system for traffickers to exploit. AI software and machine learning are currently being explored to overcome these issues through automatic identification and classification of contraband. AI programs have been developed specifically for the detection of wildlife contraband in luggage and cargo based on x-ray and CT imaging. Project SEEKER enables automated screening of 3D images to detect the presence of animal products in luggage [113]. Similarly, Project Vikela screens 2D radiological images of luggage and cargo for the automated detection of rhino horn, a highly valued and trafficked commodity from Africa [114], and can be implemented in African regions where 3D technology is not yet available. Australian researchers have also trialled the implementation of innovative 3D Real Time Tomography with an algorithm for the detection and identification of wildlife products in mail and luggage [15]. AI can reduce human error and false alarm rates and therefore improve the efficiency and accuracy of threat detection. Some AI programs will be offered free of charge to authorities once developed [60], providing a great option for under-resourced areas, but they may be difficult to implement where there are issues in the digitalization of monitoring infrastructures or outdated scanners. Although not yet widely available, the combination of AI software with already implemented ionising radiation apparatus provides a powerful tool for wildlife trafficking detection.

## Screening of persons concealing wildlife products through airports

Wildlife traffickers have been known to transport animals and wildlife products directly on their persons through airports. Non-ionising radiation techniques are ideal for use in these situations, where millimetre wave and terahertz technology are already widely available in airports and mailrooms for the detection of concealed threats. While they are not specifically designed to detect wildlife products, the literature demonstrates how full body scanners detect irregularities and could therefore be adapted to detect various instances of contraband smuggling [46,54]. While this technology would require modifications such as AI and additional testing to ensure their effectiveness, any irregularities are confirmed by a trained officer through a physical search. The difficulty with this method in relation to the wildlife trade however is that the trade can take on many different shapes and forms, therefore it would be difficult to standardise and train the image recognition software, compared with the standard shape of a gun, for example. Infrared imaging relies on the difference between body temperature and the temperature of the concealed object. This technology could be used to detect wildlife products smuggled on persons, for example by detecting body heat signatures of live animals, but would be ineffective where temperature variations are minimal. Concurrent use of infrared with terahertz or millimeter wave imaging would provide superior insight into concealed objects and overcome some of the individual limitations associated with image resolution [86]. There have been multiple instances of wild bird eggs being concealed on traffickers as they pass through airports [115–117]. While these cases were fortunately detected by security, not all persons are screened thoroughly and could easily evade detection. The use of full-body scanners on these passengers would have revealed an abnormality, warranting further investigation. These non-ionising methods typically require trained human operators to interpret images, however investment in image fusion technology and automated detection software, similar to those already being developed for other contraband types [47,80,81,94,118], could enhance their reliability.

## Screening large volumes transiting via cargo containers

The substantial volume of goods transiting daily via cargo containers poses a huge challenge for authorities as less than 2% of all containers are currently able to be inspected. Traffickers

heavily exploit these routes for the distribution of large quantities of lucrative products including pangolin scales, rhino horn, ivory and rosewood [119]. A screening tool which is already routinely being used in other ports is and currently being explored for shipping ports is detection animals. Detector dogs have already been successfully trained to detect commonly trafficked wildlife contraband, such as ivory, rhino horn and pangolin scales, and are employed at some border checkpoints [14,120–124]. The Remote Air Sampling for Canine Olfaction (RASCO) detection system is one example of a successful detection dog program which has been deployed to detect the presence of wildlife contraband in cargo containers through non-invasive canine examination of air samples [125,126]. Despite small-scale success, these animals are not being widely used in shipping ports. Meanwhile, other animal models have also shown promise for wildlife trafficking detection, where African giant pouched rats have been trained to detect hardwood timber and pangolin parts in shipments between Africa and Asia [127]. Detection animals are highly sensitive and specific in their operation and can be trained to detect a variety of different wildlife scents. Despite their accuracy however there are limitations with the implementation of detection animals, largely associated with the investment in training required and constraints of use under certain environmental or operational conditions. For example, overheating and excessive panting can reduce dog and handler performance [128], while the sensitivity of canine detection may be lowered by strong-smelling obfuscation items [14]. Where there may be hesitancy to invest in detection animals, the device equivalent, the electronic nose (eNose), has also been explored for wildlife trafficking detection. Volatilomes of wildlife and confiscated products have been characterised using 2D gas chromatography coupled with time-of-flight mass spectrometry, therefore it is possible to apply VOC detection methodology to the development of wildlife screening tools [12,129] and for the identification of unknown wildlife contraband [12,13]. This technology could provide a cost-effective, easy to use and automated alternative to detect wildlife odours in environments which are not conducive to the presence of detection animals. The concept of eNose for wildlife trafficking detection is currently being investigated [130], but not in the context of shipping containers. The device would require prior identification of specific biomarkers associated with wildlife products of interest obtained through mass spectrometry methodology. This device, once calibrated, could provide a rapid, easy to use, portable method of wildlife identification. In addition to screening at the port, smart containers are an emerging technology which record non-specific indications of illicit activity. This technology could be adapted to indirectly monitor and prevent wildlife trafficking. Smart containers are typically equipped with GPS and tampering notification systems, allowing them to be tracked throughout their journey. Therefore, any transshipping, unexpected stops, unauthorised opening of containers, or tampering of contents is recorded and used to alert authorities in real-time. This technology is already in use in many major companies and could be combined with other technologies to enhance detection capabilities, but it's effectiveness would rely on widespread adaptation across all containers and cooperation with relevant law enforcement agencies and shipping companies. The information obtained from these containers would also need to be viewed by port authorities and container operators with consideration for known trafficking routes and other risk indicators [119]. For example, if there was tampering of a container transiting from Nigeria to Vietnam, a known high-risk trafficking route for pangolin scales, the shipping documentation could be reviewed and the container inspected before it is released. Naturally, the concern with such a system is that traffickers are likely to choose containers which are not fitted with this technology, thus it may not be as useful until there is widespread adoption of the containers.

## Screening large volumes of small parcels through postal services

Live animals and their derivatives are frequently trafficked through the post. This is a particularly prevalent issue in Australia as their native reptiles are highly sought-after in the international exotic pet trade [131]. An example of a trafficking attempt was uncovered in 2021, wherein blue-tongue lizards (*Tiliqua nigrolutea*) were found bound and hidden inside DVD players at an Australian mail sorting facility. This information was obtained as part of a 15-month investigation and the package likely would not have been detected otherwise [132]. X-ray and CT scanners could have uncovered these animals as ionising radiation is particularly useful for highlighting skeletal structures. Pirotta et al. have developed an algorithm to accompany Real Time Tomography 3D X-ray imaging to detect native Australian wildlife trafficked through parcels, demonstrating the feasibility of this method [15]. However, due to the large volume of letters and parcels processed daily, it is unlikely that these scanners would be efficient enough to screen all incoming mail as a standalone detection method, even with AI. Instead, active or passive surveillance methods should first be employed to isolate dubious postal items which warrant further investigation. As previously described, dogs have been successfully trained to detect wildlife-specific scents, including reptiles [133,134]. To overcome the challenges of screening items individually, dogs could actively screen large areas efficiently and indicate the presence of wildlife products in the mail sorting room itself. This method however may be resource intensive, potentially disruptive and time consuming, especially in large facilities. Alternatively, passive surveillance via continuous air sampling could be considered. This would involve installing a device which periodically extracts air samples from a mail triage room to notify of the detection of trace VOCs or airborne biomarkers predetermined by mass spectrometry techniques, similar to an eNose device [130]. Where there is positive identification, more targeted detection protocols could be initiated to isolate the threat. The combination of large-scale screening followed by individual parcel investigation would exponentially increase detection capabilities. The proposed proactive screening approach would also need to be reflected in accompanying protocols, as screening is typically directed towards biosecurity threats entering the country rather than animal products leaving the country [135]. Thus, not only should the tools be improved, but protocols need to be adapted to prevent the movement of wildlife in and out of the country. The laws surrounding the protection of these native species are much stricter in Australia, but the same protections do not apply in the importing countries. Therefore, in the case of live animals, a proactive screening approach is also essential to safeguard animal welfare.

## Complementary screening protocols

Screening tools play a pivotal role in enhancing detection capabilities at ports, but they must be integrated with a comprehensive set of complementary components to establish a successful strategy. Effective detection achieved through a risk-based assessment is crucial for optimizing resource allocation, especially in high-traffic port environments. This approach hinges on robust communication, coordination and collaboration among port personnel, customs authorities and law enforcement agencies. Sharing intelligence and seizure data is imperative to fine-tune risk assessment algorithms and create a unified global network to combat criminal organizations. The process of gathering data relies on adequately training frontline staff and raising awareness among authorities regarding the modus operandi of illegal wildlife traffickers. This includes knowledge of how to identify illicit products, recognize suspicious behaviours and properly collect and process samples for further investigation. For example, goods originating from specific regions may require different screening protocols based on their assessed risks. Developing and maintaining an informative tool for this

purpose would necessitate collaborative efforts among entry and exit port authorities. Seizure data collected should be continually integrated into the system to enhance its effectiveness. Furthermore, countries of origin should bolster their efforts to screen outgoing cargo rather than solely relying on detection and interception at the final destination, particularly in cases where screening tools are known to be limited. Additionally, the establishment of baseline screening standards across international ports is vital to diminish opportunities for wildlife traffickers to exploit routes perceived as having weaker enforcement. The combination of screening standards and collaboration between frontline and law enforcement personnel, coupled with the targeted use of screening tools appropriate to the port requirements and suspected wildlife products, is essential to the establishment of an effective screening program for the illegal wildlife trade.

## Summary and recommendations

The illegal wildlife trade is one of the largest and most lucrative organised crimes, however publications detailing wildlife detection methods are scarce as this remains an under-researched area. There are several detection tools either currently in use or under investigation for other contraband types, some of which may be adapted for the detection of wildlife. Techniques including x-ray, CT, detection animals and electronic nose have already been used to detect wildlife at international ports and should continue to be used in conjunction with AI. Other techniques which show promise include non-ionising radiation scanners, smart containers and trace detection techniques. It is further evident that certain techniques, particularly those related to the detection of radioactive isotopes or nuclear materials, would not be appropriate for wildlife. There is likely not one solution or tool which can be universally applied, but a combination of methods may be required to enhance detection capacity, dependent on known risk factors, transport routes and composition of wildlife products traded. Physical detection tools should also be coupled with a risk-based approach to screening, particularly where high volumes of goods are involved, to enable the adequate allocation of resources to optimise performance. Efforts and funding should be directed towards developing regions facing resource limitations in particular, as these ports are at a greater risk of exploitation by wildlife traffickers and facilitate global trade. There is a dire need to strengthen our borders from the threats posed by the illegal wildlife trade and this should be reflected in the funding available for this research. Proactive research and active surveillance are essential to safeguard biodiversity, animal welfare, biosecurity and public health.

## Supporting information

**S1 Checklist. Preferred Reporting Items for Systematic reviews and Meta-Analyses extension for Scoping Reviews (PRISMA-ScR) checklist.**
(PDF)

**S1 Table. Search strategy for wildlife detection tools.**
(DOCX)

**S2 Table. Search strategy for screening tools for illicit contraband detection.**
(DOCX)

**S3 Table. Ionising radiation detection tools.** Detection tools described in the literature which primarily utilise ionising radiation.
(DOCX)

**S4 Table. Photon and neutron detection tools.** Detection tools described in the literature which primarily utilise photon and neutron interrogation techniques.
(DOCX)

**S5 Table. Nuclear detection tools.** Detection tools described in the literature which primarily utilise nuclear material detection systems.
(DOCX)

**S6 Table. Non-ionising detection tools.** Detection tools described in the literature which primarily utilise non-ionising radiation.
(DOCX)

**S7 Table. Trace detection techniques.** Detection tools described in the literature which primarily utilise trace detection techniques.
(DOCX)

**S8 Table. Acoustic and other detection tools.** Detection tools described in the literature which primarily utilise acoustic or other methods.
(DOCX)

## Acknowledgments

This review is to contribute towards the Doctor of Philosophy for G.K.M. We would like to acknowledge Associate Professor Dr Tony Hooker for his contribution to the editing of the first draft.

## Author Contributions

**Conceptualization:** Georgia Kate Moloney, Anne-Lise Chaber.

**Data curation:** Georgia Kate Moloney.

**Formal analysis:** Georgia Kate Moloney.

**Funding acquisition:** Anne-Lise Chaber.

**Investigation:** Georgia Kate Moloney, Anne-Lise Chaber.

**Methodology:** Georgia Kate Moloney, Anne-Lise Chaber.

**Supervision:** Anne-Lise Chaber.

**Visualization:** Georgia Kate Moloney.

**Writing – original draft:** Georgia Kate Moloney.

**Writing – review & editing:** Georgia Kate Moloney, Anne-Lise Chaber.

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
