## [Decision Letter · Decision Letter 0]

19 Sep 2023

PONE-D-23-23299Screening tools to detect illicit contraband at international borders and their adaptability for illegal wildlife trafficking: A scoping reviewPLOS ONE

Dear Dr. Moloney,

Thank you for submitting your manuscript to PLOS ONE. After careful consideration, we feel that it has merit but does not fully meet PLOS ONE’s publication criteria as it currently stands. Therefore, we invite you to submit a revised version of the manuscript that addresses the points raised during the review process.

We look forward to receiving your revised manuscript.

Kind regards,

Tommaso Lomonaco, Ph.D

Academic Editor

PLOS ONE

Journal Requirements:

3. Please expand the acronym “WWF and CMA CGM” (as indicated in your financial disclosure) so that it states the name of your funders in full.

4. We noted in your submission details that a portion of your manuscript may have been presented or published elsewhere. "Results from this study were published as a report with WWF. We believe it is important to publish the complete set of results in a scientific format, however, to reach the wider scientific audience." Please clarify whether this conference proceeding or publication was peer-reviewed and formally published. If this work was previously peer-reviewed and published, in the cover letter please provide the reason that this work does not constitute dual publication and should be included in the current manuscript.

7. Please include your tables as part of your main manuscript and remove the individual files. Please note that supplementary tables (should remain/ be uploaded) as separate "supporting information" files

Additional Editor Comments:

Dear authors, according to the reviewers comments I suggest major revisions.

Tommaso Lomonaco

Reviewers' comments:

Reviewer's Responses to Questions

**Comments to the Author**

1. Is the manuscript technically sound, and do the data support the conclusions?

Reviewer #1: Partly

Reviewer #2: Partly

2. Has the statistical analysis been performed appropriately and rigorously? 

Reviewer #1: N/A

Reviewer #2: N/A

3. Have the authors made all data underlying the findings in their manuscript fully available?

Reviewer #1: Yes

Reviewer #2: Yes

4. Is the manuscript presented in an intelligible fashion and written in standard English?

Reviewer #1: Yes

Reviewer #2: Yes

5. Review Comments to the Author

Reviewer #1: Please see attached comments.

The illegal movement of wildlife poses a public health, conservation and biosecurity

threat, however there are currently minimal screening tools available at international

ports of entry to intercept trafficking efforts. This review first aimed to explore the

screening tools currently available or under development for the detection of concealed

wildlife contraband at international ports, including postal services, airlines, road border

crossings and maritime routes. Where evidence was deficient, publications detailing

the use of methods to uncover other illicit substances, such as narcotics, weapons,

human trafficking, explosives, radioactive materials or special nuclear material, were

compiled and assessed for their applicability to the detection of wildlife. The first search

identified only four citations related to the detection of wildlife, however the secondary

search revealed 145 publications, including 59 journal articles and 86 conference

proceedings, describing screening tools for non-wildlife illicit contraband detection. The

screening tools uncovered were analysed to evaluate their feasibility, ease of use, and

potential applicability to wildlife contraband detection. The deficiencies evident in terms

of resource availability and research efforts targeting wildlife trafficking highlights a

potentially substantial national and international security threat which must be

addressed.

Reviewer #2: General opinion

1) The topic studied in the paper is for sure very interesting and important for the aim of fighting the illicit trafficking of wildlife, and covers undoubtfully an empty space in the literature. At the same time, the very high potentiality of this issue is not followed, at least in my opinion, by a rigorous and solid analysis of the empirical data and insights. This is a clear problem of the paper that undermine its potential role in the literature, reducing its potential usefulness for practitioners and LEAs that can improve their strategies and activities while reading it.

2) For what concerns the methodology implemented in the analysis, I have nothing to say. The methods are clearly explained and the reader is easily led through the techniques and the literature review used for the analysis.

3) The most important problem, I think, regards the very large amount of information and details that is delivered to the reader. There are too many details, the paper misses a high-level analysis and the capacity to send back a general and high-level framework.

4) The reader is trapped in a overwhelming amount of data and details; this is really preventing the capacity of promoting a coherent and robust analysis and conceptualisation of these different methods at the light of their potential, opportunities, challenges for those countries that should used them. This last issue, in particular, is completely neglected. No consideration is made on the efforts these countries should do for applying these different methods, even more if we consider that they are mainly developing countries in Africa, Asia, and Latin America that have numerous challenges to cope with, structural weaknesses, and so on.

5) The conceptual framework is weak and not entirely elaborated, so that the list of the different methods for detecting wildlife goods results to be a little bit a standing alone piece without any connection or reference to the broader literature or other conceptual framework. This is also undermining the quality of the paper and its capacity to find its placement into the broader literature on the topic of the fight against illicit wildlife trade. For example, the authors never specify the audience of the paper, so that it is a bit difficult if it more directed to LEAs and practitioners or scholars / academics. Both these categories can benefit from this kind of assessment, but they can be clearly interested in different elements or components.

Line by Line points:

Introduction:

Line 36 – road border crossing (i.e., via trucks) > only trucks, no role for cars, motorbikes, and even via foot? Check literature

Lines 61 and 62 – “... these capabilities across sectors....” and “...collaboration between sectors...” / it is not entirely clear which sectors are considered here.

Lines 71 – “Where deficiencies are evident....” > evident for what? For which kind of criteria?

Methods

From Line 77 to Line 79: can you spend few words more to explain what do these two methodologies for Scoping reviews mean for your analysis? It can be important to know a bit better how these points help frame your analysis.

Line 102 / 103: “For databases which generated thousands of results, 103 the a priori decision was made to only screen the first 200 articles” > please, justify why you have done that, how you make sure you are not excluding relevant articles or papers, and which potential consequences this can have for your empirical results

Results

I find this part weak, given that all the information is simply demanded to the Table 1 and 2; I don’t think this is a good strategy, so please anticipate at least the more relevant details / macro results in the text, so adding substance and content to a section that is, as it is now, very weak and not satisfactory enough. At the same time, the categories presented in the “Discussion” section should be moved under the “result” section; I am convinced you have to rewrite the “Discussion” section with a more high-level / summarising discussion of what your empirical results mean for the contrast / fight to the mechanisms of illicit wildlife trade for animals or plants, also considering what should be done to expand the use of tools developed for other goods but that can fit with the fight against IWT.

Line 151 / 152: “divided across six categories based on their mode of action or required sample type”

> have you elaborated these six categories or are they pre-existing categories already presented in the literature?

> at least, give us an idea of statistically relevant are the different six categories, if they have been developed for wildlife goods or for other types of trafficked goods; I mean, more substance for this section is of course very welcomed.

Table 1:

A) The lengths of the table in the annex strongly calls into question, at least from my point of view, the usefulness of this tool for summarising the empirical results. It looks to me that it can be much better to split the different macro-categories in different tables.

B) Second, please, anticipate and present some of these results already in the text; I think that the decision to leave all the explanations to the table is not so effective for the understandability of the paper and its empirical results.

C) It does not stand clearly, for me, what are those methods and tools developed specifically for wildlife, and what are instead those that have been taken by other goods and types of trafficking. This is a kind of information that should be discussed in this section.

Line 163: “Desirable” in which terms and for whom? Are we speaking about LEAs at the border or port posts (i.e. the users) or those financing the purchase of these tools? Can you give a bit more substance here?

Table 2:

A) As for Table 1, I think that a initial explanation summing up of the results presented there can be very beneficial for the potability of the empirical results; in this sense, the paper can become much more understandable and useful to study / read if you anticipate the presentation of the results in the text rather than leaving everything in the table 2.

Discussion

A) Premises: as I have already underlined in the previous paragraph about “results”, I think that many of the information you are giving in this section would fit much more under the “result” section. What is really missing here is a high-level summarisation of the results and the discussion of their relevance for what concern the strategies to fight wildlife trade. This is a key weakness of this paper.

B) First impression: too many details that could be easily optimized, not all these details are really useful for the reader; I think that the authors fall down in the trap of explaining everything with a lot of details while missing the capacity to summarising and giving explanation at high level > overwhelming the reader with not so interesting and technical information. What stands out reading the different paragraph and subparagraph of the discussion section is more a detailed list of information on the different technologies and tools rather than an analysis of the opportunities and challenges in using for fighting the illicit wildlife trade; I suggest to reduce a bit the technical part and increase in parallel the analytical one.

C) At the same time, it is not clear if these tools, when developed for other goods, are ready to be used for wildlife or need a stage of adaptation or any modification for making their use much more efficient and effective;

D) Another point that is a bit neglected is to speak about which of these tools have a real potential for innovation, where they can be combined with AI tools / digitalisation processes, and what this means for political systems that have evident issues with the digitalisation of their monitoring infrastructures.

6. PLOS authors have the option to publish the peer review history of their article (what does this mean?). If published, this will include your full peer review and any attached files.

Reviewer #1: **Yes: **Timothy C. Haas

Reviewer #2: No

---

## [Author Response · Author response to Decision Letter 0]

5 Dec 2023

Response to the Reviewers 02/11/2023 (please also see word document version attached to submission)

We would like to thank both reviewers for their positive response to our manuscript entitled ‘Screening tools to detect illicit contraband at international borders and their adaptability for illegal wildlife trafficking: A scoping review’ (now ‘Where are you hiding the pangolins? Screening tools to detect illicit contraband at international borders and their adaptability for illegal wildlife trafficking’). The comments provided by the reviewers were both insightful and helpful in improving the quality of our manuscript. We have addressed each comment provided by the reviewers individually, with the original comment or question in italics followed by our response. Please note that all line numbers referenced in our responses refer to those when track changes are not visible (disabled).

Reviewer 1

Thank you for your comments and editing suggestions. We would like to address your suggestions in the order they were provided:

1. There is too much detail on methods that are inapplicable to wildlife screening. Those methods need to be removed from the manuscript. 

We reviewed the methods and shortened the text where we felt was appropriate. We did not however reduce the methods significantly at this stage, as we believe the explanation provided is important for the readers to understand how we conducted the review. The second reviewer also provided positive feedback for the methods, so the conflicting feedback made us reluctant to make any further changes at this stage.

2. Most statements about the potential application of a method to wildlife detection are too vague to be useful. These descriptions need to be enlarged to include details as to exactly what modifications would be needed before a method could be used to detect wildlife in luggage or in shipping containers. For example, lines 461-464 are a good start but need much more detail.

Thank you for your insight. Upon review, we agree that the details provided in the first version of the manuscript were too vague. We have instead decided to take a new approach, whereby we have presented a realistic case-based scenario for each port of operation and how particular tools may help in that circumstance. We hope that this approach presents a clearer discussion and, coupled with the revised tables in the results, provides a more useful manuscript overall. 

3. There is little discussion of the challenges of applying a particular method to detecting wildlife within a particular setting such as airport passenger screening versus shipping container screening. 

We hope that we have now addressed this concern throughout the discussion, where we have addressed each port environment separately with reference to how screening tools may be applied in each scenario for wildlife trafficking detection. 

4. The manuscript’s title in misleading. Actually, there is little in the way of a literature review. Most of the manuscript contains very brief descriptions of screening methods for detecting narcotics, explosives, or firearms. Similar descriptions can easily be found elsewhere. Hence, these descriptions need to be considerably abbreviated or, in many cases, removed entirely.

We have restructured the manuscript considerably and have restricted the detection tools described in any detail to those which could be applied to wildlife trafficking. We hope that the reviewer will find the revised version of the manuscript more conducive to a scoping review, which was the author’s intention. We have also adjusted the title slightly to better convey the outcome of the manuscript. 

Reviewer 2:

Thank you for your time and comments. We would like to address the concerns raised in order:

1) The topic studied in the paper is for sure very interesting and important for the aim of fighting the illicit trafficking of wildlife, and covers undoubtfully an empty space in the literature. At the same time, the very high potentiality of this issue is not followed, at least in my opinion, by a rigorous and solid analysis of the empirical data and insights. This is a clear problem of the paper that undermine its potential role in the literature, reducing its potential usefulness for practitioners and LEAs that can improve their strategies and activities while reading it.

Thank you for your insight. We have restructured the manuscript significantly in an attempt to make this resource more useful for law enforcement agencies and customs authorities. We hope that the reviewers will agree with the changes made.

2) For what concerns the methodology implemented in the analysis, I have nothing to say. The methods are clearly explained and the reader is easily led through the techniques and the literature review used for the analysis.

Thank you for your comments regarding the methods.

3) The most important problem, I think, regards the very large amount of information and details that is delivered to the reader. There are too many details, the paper misses a high-level analysis and the capacity to send back a general and high-level framework.

We have reviewed the manuscript and agree with the reviewer. We have instead decided to restrict the information presented largely to the tools which are deemed adaptable for wildlife trafficking and have moved the additional tools to the supplementary information. We hope that our new approach will make this a more useful and readable guide for law enforcement personnel and will not overwhelm our readers.

 

4) The reader is trapped in an overwhelming amount of data and details; this is really preventing the capacity of promoting a coherent and robust analysis and conceptualisation of these different methods at the light of their potential, opportunities, challenges for those countries that should used them. This last issue, in particular, is completely neglected. No consideration is made on the efforts these countries should do for applying these different methods, even more if we consider that they are mainly developing countries in Africa, Asia, and Latin America that have numerous challenges to cope with, structural weaknesses, and so on.

Thank you for your feedback. We have taken your comments into account and have made efforts to address these issues. In our revised analysis, we have streamlined the presentation of data and details to ensure a more coherent and focused discussion of the various methods. We understand the importance of promoting a robust analysis and conceptualization of these methods, considering their potential, opportunities, and challenges for the countries in question. We have also taken your point about developing countries in Africa, Asia, and Latin America into consideration.

In the updated version, we have included notes and comments dedicated to addressing the efforts and challenges faced by developing countries in implementing these methods. We hope that these revisions address your concerns and provide a more balanced and informative analysis. Thank you again for your valuable feedback.

5) The conceptual framework is weak and not entirely elaborated, so that the list of the different methods for detecting wildlife goods results to be a little bit a standing alone piece without any connection or reference to the broader literature or other conceptual framework. This is also undermining the quality of the paper and its capacity to find its placement into the broader literature on the topic of the fight against illicit wildlife trade. For example, the authors never specify the audience of the paper, so that it is a bit difficult if it more directed to LEAs and practitioners or scholars / academics. Both these categories can benefit from this kind of assessment, but they can be clearly interested in different elements or components.

Upon review, we agree with the reviewer regarding the structure of the manuscript. We are primarily aiming to provide a guide for law enforcement authorities, as now stated in the introduction (lines 71-72). We have tried to better incorporate the different types of tools available with case-based scenarios which might be encountered, with reference to particular challenges faced by personnel at each port type. 

Line by Line points:

Introduction:

Line 36 – road border crossing (i.e., via trucks) > only trucks, no role for cars, motorbikes, and even via foot? Check literature

Road border crossing can include trucks, cars, motorbikes, and on foot. “Trucks” was only an example, which we have now removed to avoid confusion.

Lines 61 and 62 – “... these capabilities across sectors....” and “...collaboration between sectors...” / it is not entirely clear which sectors are considered here.

The sectors considered here include those highlighted as part of the ROUTES partnership and IMO guidelines, “transport and logistics companies, government agencies, development groups, law enforcement, conservation organizations, academia” (line 42-43). We wanted to highlight how these initiatives demonstrate the need for collaboration between LEAs, transport sectors (postal, maritime, airlines) and others are required for a successful wildlife trafficking detection program. 

Lines 71 – “Where deficiencies are evident....” > evident for what? For which kind of criteria?

We have updated this line for clarity (lines 68-71).

Methods

From Line 77 to Line 79: can you spend few words more to explain what do these two methodologies for Scoping reviews mean for your analysis? It can be important to know a bit better how these points help frame your analysis.

The methodologies offer a standardized approach to scoping reviews, which means that analysis is thorough and well-informed (lines 76-79).

Line 102 / 103: “For databases which generated thousands of results, 103 the a priori decision was made to only screen the first 200 articles” > please, justify why you have done that, how you make sure you are not excluding relevant articles or papers, and which potential consequences this can have for your empirical results

While this approach expedites the initial screening process, we acknowledge that it may have potential consequences for our empirical results:

1. Risk of Missing Relevant Articles: By limiting the number of articles screened, we might miss some relevant papers buried deeper in the search results. However, this limitation is balanced by our subsequent rigorous screening and selection processes.

2. Potential for Bias: There's a risk of bias in the results, as the first 200 articles might not be entirely representative of the entire body of literature. This is a trade-off we considered in light of resource constraints and time limitations.

We have included a statement encompassing these considerations (line 101-105).

Results

I find this part weak, given that all the information is simply demanded to the Table 1 and 2; I don’t think this is a good strategy, so please anticipate at least the more relevant details / macro results in the text, so adding substance and content to a section that is, as it is now, very weak and not satisfactory enough. At the same time, the categories presented in the “Discussion” section should be moved under the “result” section; I am convinced you have to rewrite the “Discussion” section with a more high-level / summarising discussion of what your empirical results mean for the contrast / fight to the mechanisms of illicit wildlife trade for animals or plants, also considering what should be done to expand the use of tools developed for other goods but that can fit with the fight against IWT.

Thank you for your comments, we have totally revamped the results and discussion sections and hopefully made it more practical for LEAs by including scenarios.

Line 151 / 152: “divided across six categories based on their mode of action or required sample type”

> have you elaborated these six categories or are they pre-existing categories already presented in the literature?

> at least, give us an idea of statistically relevant are the different six categories, if they have been developed for wildlife goods or for other types of trafficked goods; I mean, more substance for this section is of course very welcomed.

Thank you, we have elaborated the 6 categories based on technical principles in the results section. It was the decision of the authors to divide the tools in this manner to make it easier for the reader to comprehend the information presented. These categories have been referred to across various publications, and while they do not represent a particular convention of naming, they do help to identify shared properties and mechanisms of action.

Table 1:

A) The lengths of the table in the annex strongly calls into question, at least from my point of view, the usefulness of this tool for summarising the empirical results. It looks to me that it can be much better to split the different macro-categories in different tables.

We have divided the tables in the supplementary information into their six categories for ease of use. We hope that this addresses the reviewer’s concern.

B) Second, please, anticipate and present some of these results already in the text; I think that the decision to leave all the explanations to the table is not so effective for the understandability of the paper and its empirical results.

We have now restructured the results section and have tried to summarise the screening tools more clearly. 

C) It does not stand clearly, for me, what are those methods and tools developed specifically for wildlife, and what are instead those that have been taken by other goods and types of trafficking. This is a kind of information that should be discussed in this section.

We have more clearly highlighted where tools have already been trialed or used for wildlife both in Table 2 and throughout the discussion. 

Line 163: “Desirable” in which terms and for whom? Are we speaking about LEAs at the border or port posts (i.e. the users) or those financing the purchase of these tools? Can you give a bit more substance here?

We have now specified that: “We aim to provide a practical guide for law enforcement agencies (LEA) to strengthen their knowledge and capabilities and to raise awareness for the significance of the illegal wildlife trade” (line 71-72). Hence, the properties outlined are deemed desirable by the users (LEAs) (see line 305-307).

Table 2:

A) As for Table 1, I think that a initial explanation summing up of the results presented there can be very beneficial for the potability of the empirical results; in this sense, the paper can become much more understandable and useful to study / read if you anticipate the presentation of the results in the text rather than leaving everything in the table 2.

We have restructured the results section and Table 1. The aim of Table 1 is to present the tools which have been, or could be, adapted for wildlife trafficking, with their benefits and limitations are outlined in the literature. These features and tools have also been summarized in the results section and in the discussion. We hope that we have addressed the reviewers concerns with these changes.

Discussion

A) Premises: as I have already underlined in the previous paragraph about “results”, I think that many of the information you are giving in this section would fit much more under the “result” section. What is really missing here is a high-level summarisation of the results and the discussion of their relevance for what concern the strategies to fight wildlife trade. This is a key weakness of this paper.

We agree and have moved the information previously constituting the discussion to the results section and have refined it considerably. 

B) First impression: too many details that could be easily optimized, not all these details are really useful for the reader; I think that the authors fall down in the trap of explaining everything with a lot of details while missing the capacity to summarising and giving explanation at high level > overwhelming the reader with not so interesting and technical information. What stands out reading the different paragraph and subparagraph of the discussion section is more a detailed list of information on the different technologies and tools rather than an analysis of the opportunities and challenges in using for fighting the illicit wildlife trade; I suggest to reduce a bit the technical part and increase in parallel the analytical one.

We agree with the reviewers assessment, and as previously stated have restructured the discussion to instead focus on how the tools identified as potentially adaptable may be used for wildlife with reference to case-based scenarios.

C) At the same time, it is not clear if these tools, when developed for other goods, are ready to be used for wildlife or need a stage of adaptation or any modification for making their use much more efficient and effective.

We hope that we have addressed this point throughout the discussion, as we have included where modifications would be required where applicable.

D) Another point that is a bit neglected is to speak about which of these tools have a real potential for innovation, where they can be combined with AI tools / digitalisation processes, and what this means for political systems that have evident issues with the digitalisation of their monitoring infrastructures.

Thank you for your insightful comment. We wholeheartedly agree with your point. We have now indeed highlighted the potential of these tools for innovation, particularly in conjunction with AI and digitalization processes.

---

## [Decision Letter · Decision Letter 1]

6 Feb 2024

Where are you hiding the pangolins? Screening tools to detect illicit contraband at international borders and their adaptability for illegal wildlife trafficking

PONE-D-23-23299R1

Dear Dr. Georgia Moloney,

We’re pleased to inform you that your manuscript has been judged scientifically suitable for publication and will be formally accepted for publication once it meets all outstanding technical requirements.

Kind regards,

Tommaso Lomonaco, Ph.D

Academic Editor

PLOS ONE

Additional Editor Comments (optional):

Reviewers' comments:

Reviewer's Responses to Questions

**Comments to the Author**

1. If the authors have adequately addressed your comments raised in a previous round of review and you feel that this manuscript is now acceptable for publication, you may indicate that here to bypass the “Comments to the Author” section, enter your conflict of interest statement in the “Confidential to Editor” section, and submit your "Accept" recommendation.

Reviewer #1: All comments have been addressed

2. Is the manuscript technically sound, and do the data support the conclusions?

Reviewer #1: Yes

3. Has the statistical analysis been performed appropriately and rigorously? 

Reviewer #1: N/A

4. Have the authors made all data underlying the findings in their manuscript fully available?

Reviewer #1: Yes

5. Is the manuscript presented in an intelligible fashion and written in standard English?

Reviewer #1: Yes

6. Review Comments to the Author

Reviewer #1: The revision has a much-improved focus on detection of illicit wildlife shipments. The balance between review of all technologies versus those with potential application to wildlife detection is right. Table 2 is excellent but it should be referred to more frequently in the Discussion so as to give the reader a way to organize the many recommendations and comments made in the Discussion. I think the article is an important contribution to the literature and points to how far we need to go before illicit wildlife shipments have any hope of being detected.

7. PLOS authors have the option to publish the peer review history of their article (what does this mean?). If published, this will include your full peer review and any attached files.

Reviewer #1: **Yes: **Timothy C. Haas

---

## [Editor Report · Acceptance letter]

25 Mar 2024

PONE-D-23-23299R1 

PLOS ONE

Dear Dr. Moloney, 

I'm pleased to inform you that your manuscript has been deemed suitable for publication in PLOS ONE. Congratulations! Your manuscript is now being handed over to our production team.

Kind regards, 

on behalf of

Dr. Tommaso Lomonaco 

Academic Editor

PLOS ONE